bioengineering/ecology/evolution

secretions, biocementation, extended phenotype, weathering resistance, ease of handling, self-weight consolidation

**Author for correspondence:**
Renee M. Borges
e-mail: renee@iisc.ac.in

# Moisture alone is sufficient to impart strength but not weathering resistance to termite mound soil

Nikita Zachariah[1], Tejas G. Murthy[2]
and Renee M. Borges[1]

[1]Centre for Ecological Sciences, and [2]Department of Civil Engineering, Indian Institute of Science, Bangalore 560012, India

 NZ, 0000-0001-9512-4513; RMB, 0000-0001-8586-7380

Soil is used for the construction of structures by many animals, at times admixed with endogenous secretions. These additives, along with soil components, are suggested to have a role in biocementation. However, the relative contribution of endogenous and exogenous materials to soil strength has not been adequately established. Termite mounds are earthen structures with exceptional strength and durability including weathering resistance to wind and rain. With *in situ* and laboratory-based experiments, we demonstrate that the fungus-farming termite *Odontotermes obesus* which builds soil nest mounds, when given a choice, prefers soil close to its liquid limit for construction. At this moisture content, the soil–water mixture alone even in the absence of termite handling undergoes self-weight consolidation and upon drying attains a monolithic, densely packed structure with compressive strength comparable to the *in situ* strength of the mound soil; however, the soil–water mixture alone has lower resistance to water erosion than the *in situ* mound samples, suggesting that termite secretions impart weathering resistance and thereby long-term stability to the mound. Therefore, weathering resistance and compressive strength are conferred by different aspects of termite soil manipulation. Our work provides novel insights into termite mound construction and strength correlates for earthen structures built by animals.

## 1. Introduction

Soil and marine sediments have been used for the construction of various structures by humans and non-human animals. These structures provide shelter [1], protection from predators [2–4], help in prey capture [5], thermoregulation [6] and mate

attraction [7,8], and act as incubators for fungus cultivation [1,9]. Different materials can be employed, e.g. underwater rock materials [10], sand [7,8] and silicic acid [11]. Most materials require some form of inter-granular cementation or strengthening of inter-granular bonds for structure stability (except for non-load-bearing structures such as antlion nests [5] and geometric circles made by puffer fish in the sand [7,8]). Various factors probably impart strength to building materials, e.g. phosphorylated H-fibroin serines in caddisfly larval silk to bind leaves, sticks or stones for building body cases [12], foam-like adhesive in the reef-building polychaete *Phragmatopoma californica* to bind particulate matter [4] and soil composition and moisture content in mud nests of swallows in addition to saliva [2]. However, the strength correlates, i.e. the relationship between material additives and strength, have not been investigated in such cases of suggested biocementation. Also, the weathering resistance of cemented terrestrial structures, e.g. termite mounds and swallow nests, awaits investigation.

Conventional soil engineering studies have shown that soil strength depends on its mineral composition [13], grain size distribution (proportion of gravel, sand silt and clay [14]), degree of saturation [15] and stress history [16]. Another important factor affecting soil strength is the presence of water in pores and the interparticle junctions of soil. The surface tension of water attracts soil particles together and imparts strength to the soil [17]. Soil suction is conceptualized as stresses arising from interparticle cementation, van der Waals attraction, double-layer repulsion and capillary stresses [18]. In practical terms, soil suction is a measure of the affinity of soil to retain water and can provide information on soil parameters that are influenced by the soil water, for example, volume change, deformation and strength characteristics of the soil [19]. Increase in soil water content reduces the soil suction—in saturated soils the suction value is zero; suction increases with a decrease in soil water content [18]. Soil suction measurements, therefore, help in establishing the contribution of water surface tension towards soil strength. Soil suction has been suggested to play an important role in tunnel excavation and tunnel stability in ants [20]. Soil strength could increase by repeated wetting and drying [21] probably caused by the rearrangement of soil structure upon wetting—the presence of moisture in soil pores may change the orientation of individual soil particles. Cementation is also important in imparting strength to the soil. Cementation can occur naturally (e.g. caused by precipitation of calcite [22], silica [23], alumina [24]) or be artificially induced (e.g. addition of agents such as cement [25,26] or lime [27] in human constructions). The challenge in animal-constructed structures made of soil or other granular materials is to determine what factors, whether purely material-based or exogenous (water or other additives), contribute to their strength and stability including resistance to weathering. Weathering is the irreversible response of soil and rocks to natural or artificial exposure to processes affecting the near-surface geomorphological environment [28]. Mechanical, chemical and biochemical processes can affect weathering of earthen structures made by animals such as termite mounds.

Termite nest mounds are iconic earthen structures. At 2.5–10 m in height, they are up to three to four orders of magnitude greater in length than individual termites [29,30], remain intact for decades [30] or even centuries [31] and are 10 times stronger than the surrounding soil [32]. Biocementation [32], matrix suction [32], clay minerals [33], termite salivary amylase [34] and polysaccharides [35] have been suggested to play a role in termite mound cementation. However, adequate tests have not been performed to elucidate their role in strength acquisition via cementation.

Major and minor termite worker castes can mould moist soil (between 15–60% moisture content percentage dry weight [30]) into aggregates called boluses (singular: bolus), that act as the basic building blocks of construction, and are analogous to bricks used in construction by humans [30,32]. While manufacturing boluses, moist soil is mixed with termite secretions (visualized by offering glass beads to termites [30]) which has been suggested to act as a biocement [32]. The nest mounds of a model mound-building termite *Odontotermes obesus* are abundant in the red soil of less than 75 μ particle size and minerals like kaolinite [32]. Soil devoid of organic matter is difficult for these termites to handle and the boluses made with this soil are very fragile suggesting that soil organic matter plays a role in termite mound cementation [30]. Also, termite mounds show considerable resistance to repeated cycles of wetting and drying [32]. While there are many types of weathering [28], the removal of soil particles from termite mounds can also occur by rainwater runoff from the outer mound walls. Therefore, the integrity of termite mounds, and the survival of the colony within, depends on the weathering resistance of the termite mound soil.

Using *O. obesus* as a model organism, this study investigates the following factors in imparting strength to termite mound soil: (i) soil cementation by termite endogenous additives, (ii) soil suction caused by surface tension of water between soil particles, and (iii) self-weight consolidation of soil. The study also quantifies the weathering resistance of termite mound soil under various moisture and temperature regimes.

# 2. Material and methods

## 2.1. Study species and site

This study was conducted on *O. obesus* termites at the Indian Institute of Science campus in Bangalore, India. *Odontotermes obesus* is widely distributed in India [36], cultivates fungus [9] and constructs mounds that can be up to 2.5 m tall [30] and that have been observed to last for more than 10 years [30]. The study region has red soil dominated by the clay minerals kaolinite and montmorillonite, and the non-clay minerals quartz, mica and feldspar [37].

## 2.2. Factors contributing to soil strength in termite mounds

In order to understand the relative contribution of soil organic matter and termite secretions towards compressive strength (referred to as 'strength' henceforth) to termite mound soil, we offered red soil and burnt soil (red soil combusted at high temperature to remove organic matter [38]) to termites in the laboratory, collected boluses, packed them in cylindrical moulds (2 cm height × 1 cm diameter), allowed them to dry under room conditions followed by drying overnight at 80°C and then tested their strength following ASTM protocol [38]. The samples were placed in a Universal Testing Machine and were subjected to unconfined compression at 1 mm min$^{-1}$ until sample failure [39]. Stress–strain graphs were plotted for each sample. The highest stress value in the graph indicated the peak compressive stress. We also compared the strength of red soil and burnt soil in the absence of handling by termites. Burnt soil samples (boluses and soil unmanipulated by termites) were extremely fragile and, upon packing in cylindrical moulds and subjected to drying, would crumble. Therefore, all further experiments with soil were conducted with red soil (soil containing organic matter). We previously established that soil samples of 2 cm height × 1 cm diameter show the same behaviour under stress as soil samples of 6 cm height × 3 cm diameter (recommended by ASTM [38]) and that no scaling of strength occurs [39]. Still, preparing cylindrical moulds with boluses required obtaining a sufficient number of boluses from termites at a very high rate. This required understanding (i) favourable soil moisture content for soil manipulation by termites because, in nature, termites are expected to manipulate soil at the most favourable moisture content in order to increase the rate of material transfer during construction, and (ii) time required for soil to dry and attain asymptotic strength. The following experiments were carried out to determine these parameters.

## 2.3. *In situ* moisture content of boluses made during breach repair

Intentional breaches were made in five termite mounds and individual freshly deposited boluses (boluses merge together after deposition [30]) were collected in pre-weighed, air-tight Eppendorf tubes, brought to the laboratory and their moisture content determined. Efforts were made to minimize moisture loss during bolus collection; however, some moisture loss was inevitable. Therefore, the values obtained for *in situ* moisture content of boluses are likely to be a slight underestimate.

## 2.4. Effect of soil moisture content on ease of handling

In order to understand the preferred moisture content for soil manipulation by termites, termites were offered less than 75 µ particle size soil in a range of moisture contents (15%, 17%, 20%, 30%, 33%, 40%, 50%, 60%) in Petri dishes (35 mm diameter). The time taken to start making boluses (latency; $T$) and the rate of bolus making were recorded (number of boluses made in 20 min from the start of bolus making) following the method of Zachariah *et al.* [30]. Ten replicate tests were separately conducted for major and minor workers. The reciprocal of latency ($1/T$) and the rate of bolus making were calculated, the values normalized to the highest values in each category and plotted. These were considered as measures of the ease of material handling by termites [30].

## 2.5. Effect of drying on soil strength

The results of experiments performed in §2.4 (see §3.2 for results) suggested that soil between 30 and 33% moisture content (per unit dry weight) was most easily handled. Therefore, less than 75 µ particle size soil (without termite manipulation) at 30% moisture content was filled in cylindrical moulds (2 cm

height × 1 cm diameter) and was allowed to dry for the different number of days under room conditions (electronic supplementary material, figure S1). The cylindrical moulds used for this and subsequent experiments were sealed at the base and the soil allowed to dry only from the top. This is unlike what happens in termite mounds where the external walls of the mound are exposed to the atmosphere and are expected to dry from all sides. However, this is unlikely to affect strength because the density achieved in the case of samples prepared in this manner (1.7 g cm$^{-3}$) was comparable to the *in situ* mound soil density reported by Kandasami *et al.* [32]. The samples were de-moulded, oven dried overnight at 80°C, tested under unconfined compression at 1 mm min$^{-1}$ displacement and peak compressive stresses were plotted. Samples could only be properly de-moulded at 9 days of drying and the experiment was carried out until 16 days of drying with six replicates for each day of drying. Air drying of soil over several days allowed sedimentation and interlocking of soil particles. Air drying of soil samples was an attempt at mimicking the natural process of drying of soil in the termite mound. The remnant moisture in the soil sample was removed by oven drying in order to ensure that it did not affect the process of compressive strength testing. This exercise could not be done with termite boluses because it was difficult to obtain the sufficient number of boluses for filling multiple moulds daily.

## 2.6. Effect of moisture content on self-weight consolidation of soil

The results from the previous experiment suggested that the peak compressive strength of soil without termite manipulation was similar to the peak compressive strength of *in situ* termite mound soil as reported by Kandasami *et al.* [32] and by Zachariah *et al.* [39]. Therefore, we tested the role of self-weight consolidation (densification of soil under its own weight without application of any external force [40]) in the strength of mound soil. For this, less than 75 µ particle size soil mixed with water (30%, 40%, 50%, 60% dry weight of soil) was filled in cylindrical moulds (2 cm height × 1 cm diameter) as before and was left open to the air for drying for 12 days (drying from the upper surface only as mentioned earlier; electronic supplementary material, figure S1). After 12 days, the samples were de-moulded. The samples were tested under unconfined compression at 1 mm min$^{-1}$ displacement. For samples at 15% and 20% moisture contents, self-weight consolidation did not yield cylindrical samples that could be de-moulded; therefore, these soil samples were compacted into moulds by physical agitation, dried under room conditions followed by oven drying and then tested as described above. The surface features of the samples (packing of soil particles inside the samples observed on the sample surface) were imaged under a light microscope before testing them under compression. The dry density was calculated for each sample as the dry density of soil is positively correlated with its peak compressive stress value [40,41].

## 2.7. Contribution of termite secretions towards strength of aggregated soil

The deposition of salivary secretions by termites was observed to cohere glass beads into glass boluses under electron microscopy [30], and these secretions were proposed to play a role in soil cementation [32]. However, the exact contribution of these secretions in soil cementation was not known. The following two experiments were performed to understand how much additional strength secretions impart to soil above that achieved by moisture-facilitated soil suction alone. Because approximately 30–33% water content was favourable for soil manipulation by termites (see Results §3.2), this moisture content was selected for further experimentation. Major and minor workers were collected from three mounds (1 : 1 ratio) and were offered less than 75 µ particle size soil at 30–33% moisture content in Petri dishes in the laboratory. The Petri dishes were sealed with Parafilm to prevent moisture loss during experimentation and termites were allowed to make boluses for 18–24 h under dark conditions. The boluses were filled in cylindrical moulds as mentioned before and were allowed to undergo self-weight consolidation. The soil in the absence of termite manipulation was allowed to undergo self-weight consolidation for the same duration in control moulds. Samples were trimmed to 1 cm × 2 cm (diameter × height) and tested under unconfined compression at 1 mm min$^{-1}$ displacement as above. The peak compressive strength of boluses and control soil samples were compared (*n* = 10 pairs).

Samples were also cored out from an abandoned termite mound used by Kandasami *et al.* [32] (i.e. slice A7), trimmed and tested under unconfined compression as mentioned above. This was done to understand the effect of stress history and wetting–drying cycles on soil strength because soil from the abandoned termite mound must have a history of many cycles of wetting, drying and manipulation by termites.

## 2.8. Total soil suction measurement

Because termite mound soil is known to contain *in situ* moisture [39], and soil alone after self-weight consolidation attains strength comparable to the *in situ* mound strength, therefore, the role of soil suction in imparting strength to the soil was investigated. Samples were obtained from live mounds and their dry density and moisture contents were determined. The average moisture content of *in situ* termite mound soil for these experimental samples was found to be 6.2% per unit dry weight of soil. Samples were also prepared with control of red soil (soil with organic matter) at 14% and 30% moisture levels. The soil samples were packed in moulds at the same dry density as the *in situ* mound soil and were demoulded and dried under room conditions until they reached approximately 6% moisture content. Equal weights of mound soil and red soil samples were placed in metal containers and total soil suction was calculated as per ASTM standards [19].

## 2.9. Weathering resistance of soil with and without termite manipulation

In nature, the weathering of soil by water is well documented. Usually termite mounds encounter repeated cycles of wetting and drying which could cause weathering. In order to quantify this, samples were extracted from an abandoned mound and several live mounds and were subjected to repeated cycles of wetting and drying. Soil samples (150–200 g) were immersed in distilled water for increasing time intervals (2, 4, 8, 16, 32, … ,min) and dried until complete sample disintegration following the protocol used by Kandasami *et al.* [32]. In this experiment, we were trying to decipher the effect of termite secretions, mound *in situ* particle arrangement and soil history (ageing effects) on weathering resistance of the soil. The following comparisons were made: (i) abandoned mound samples stored in the laboratory for 6 years (slice A7 from Kandasami *et al.* [32]) (effect of termite secretions, mound *in situ* particle arrangement and soil history), (ii) samples from occupied mounds dried at 80°C initially and also after every cycle of wetting (effect of termite secretions and mound *in situ* particle arrangement), (iii) samples from occupied mounds not dried initially but dried at 80°C after every cycle of wetting (same effects as above), (iv) samples from occupied mounds not dried initially but dried at 37.5°C after every cycle of wetting (same effects as above), (v) mound soil crushed and reconstituted at 17% moisture content at *in situ* dry density (effect of termite secretions on reconstituted mound soil), (vi) mound soil crushed and reconstituted at 30% moisture content and allowed self-weight consolidation (same effects as above), and (vii) control soil collected from 1–2 feet below the ground near termite mounds, mixed with 30% water and allowed to undergo self-weight consolidation (effect of mound *in situ* particle arrangement and soil history without termite secretions). The moisture levels of 17% and 30% were chosen for the above experiment as Kandasami *et al.* [32] reported *in situ* soil moisture after breach repair to be approximately 17%, and the present study shows that the moisture content of fresh boluses at the mound is approximately 30%. The drying temperature of 80°C was selected to ensure complete drying of samples, whereas drying at 37.5°C was close to the maximum temperature recorded in the study site at Bangalore and was intended to mimic natural drying temperatures. Six replicates were used in each case. Some samples were dried before subjecting them to wetting (as mentioned above) in order to ensure that any weight loss can be attributed to weathering alone. Samples from occupied mounds (type (iv) mentioned above) were not dried initially to understand the role of *in situ* moisture (which is about 6%; see results) in soil weathering. Completely immersing samples in water tests an extreme condition of weathering, because in nature, termite mounds are rarely submerged in water.

## 2.10. Statistical analysis

We analysed the data using the software package R version 3.5.1 (2018–07–02). Data were tested for normality using the Shapiro–Wilk test. For data on the effect of drying on soil strength, an analysis of variance was performed with aov function: peak compressive stress ∼ days of drying, followed by Tukey's HSD post-hoc tests. For data on the effect of moisture content on soil strength, an analysis of variance was employed with aov function: peak compressive stress ∼ initial soil moisture content, separately for compacted soil and for self-weight consolidated soil followed by Tukey's HSD post-hoc tests. For data on the contribution of termite secretion towards soil strength, unpaired Wilcoxon signed-rank tests were employed for comparing the strength of boluses made in the laboratory versus control soil, *in situ* mound samples versus control soil and termite boluses versus *in situ* mound samples.

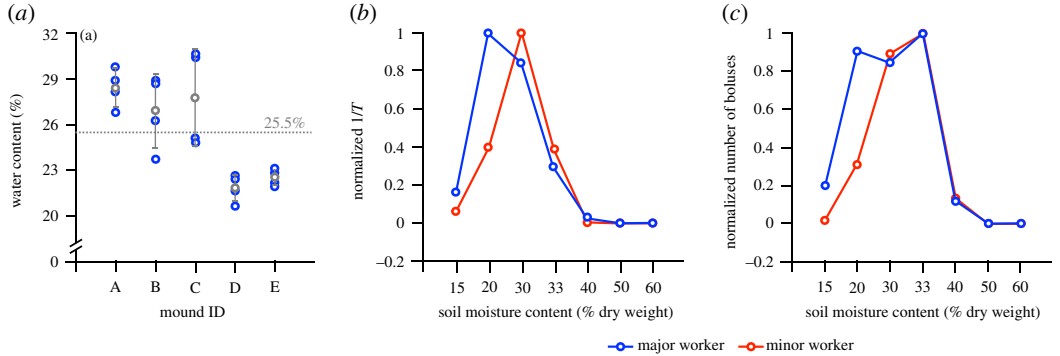

**Figure 1.** Soil moisture preference of termites. (*a*) *In situ* moisture content of mound soil during breach repair. Blue circles represent moisture content of individual samples, grey circles represent averages and error bars represent standard deviations. Dashed line represents the overall average, $n = 4$ for all mounds. (*b,c*) Ease of handling of less than 75 µ soil particle size at different moisture contents (latency and number of boluses carried in a fixed time interval; values normalized to the highest value in each category). Values represent averages for 10 replicates.

# 3. Results and discussion

## 3.1. *In situ* moisture content of soil boluses during breach repair

*In situ* moisture content of soil boluses carried by termites during breach repair was 20–31% (per cent dry weight) (average 25.5%; figure 1*a*). Among the physical characteristics reported for soils, liquid and plastic limits are often reported. The plastic limit is defined as the water content (in percentage) of soil at the boundary between the plastic and the semi-solid states [42]. Liquid limit is the water content, in percentage, of soil at the arbitrarily defined boundary between the semi-liquid and plastic states [42]. In the range of water content between the plastic and the liquid limits, soils can be easily deformed and moulded [42,43]. The plastic and liquid limits of *O. obesus* mound soils in this study region lie between 16.9–18.3% and 33.7–36.2% moisture per unit dry weight, respectively [32]. In this study, the *in situ* moisture content of soil boluses prepared by termites was within the plastic region of the soil (which could allow self-weight consolidation of soil) (self-weight of soil in this case). Therefore, in termite mounds, the boluses deposited during construction coalesce with each other owing to their moisture content and become individually indistinguishable [30]. The formation of a uniform soil mass from the coalescence of boluses has implications for the strength and ventilation of termite mounds [39].

## 3.2. Effect of soil moisture content on ease of handling

Highest ease of handling was observed around 30–33% moisture content for both major and minor workers (figure 1*b,c*). This ease of handling was 5–10 times higher than that at other moisture contents. Ease of handling was lower for the soil of both higher and lower moisture contents probably because, at lower moisture contents, it is difficult for termites to adhere soil particles together with their secretions alone for making boluses, and at higher moisture contents the soil resembles a slurry and can affect the rate of movement of termites and their bolus-making abilities (we earlier reported that the rate of material handling by termites is reduced in materials of very low rigidity [30]).

## 3.3. Effect of drying on soil strength

The peak compressive stress continued to increase with the time of drying ($F_{1,31} = 3.46$, $p < 0.01$) and attained the highest value around 12 days when it was comparable to the *in situ* mound soil strength (figure 2; electronic supplementary material, table S1) [32,39]. During these 12 days, the sedimentation of soil particles and evaporation of water from the interparticle spaces would have led to the interlocking of soil particles and the formation of capillary bridges between the soil particles. For all further experiments, the samples were dried for 12 days in order to ensure that they attain their peak strength under the experimental conditions. Drying under room conditions resembles drying at the outer walls of the termite mound which are in contact with the atmosphere. The interior of the

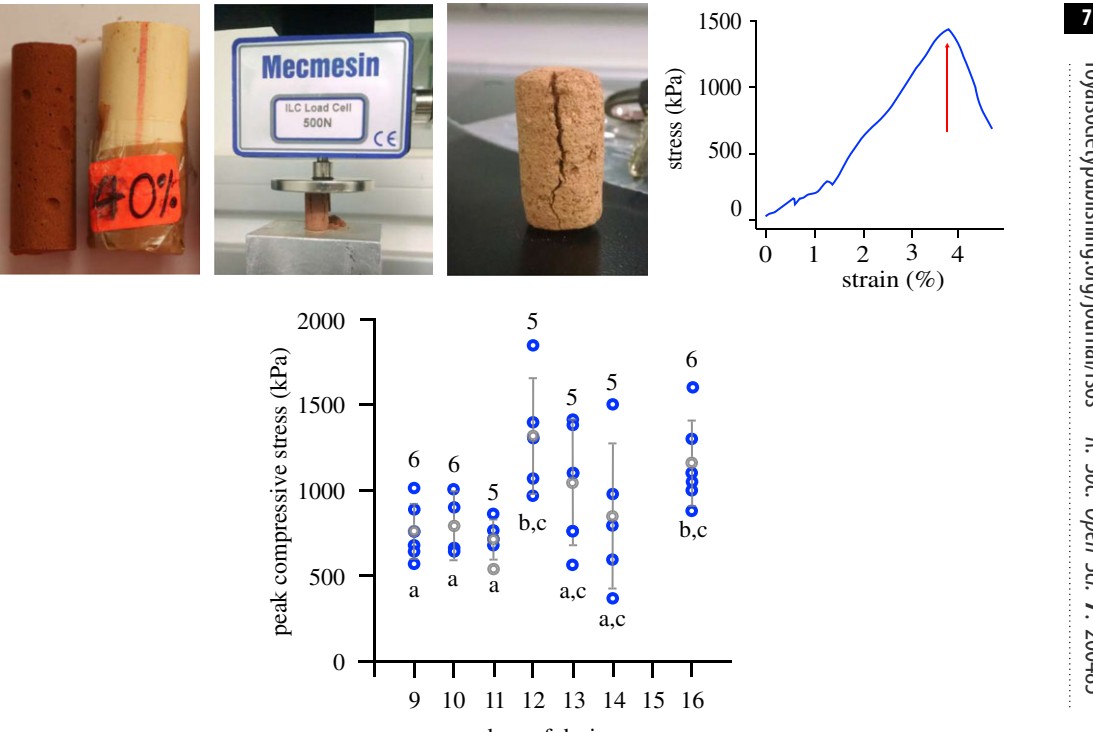

**Figure 2.** Change in peak compressive stress during drying of less than 75 μ particle size red soil at initial moisture content of 30% (per cent dry weight of soil). Blue circles indicate peak compressive stress of individual samples, grey circles indicate means, and error bars indicate standard deviations. Numbers above circles indicates sample sizes. Letters indicate significant differences obtained from Tukey's HSD tests in comparison between samples with different days of drying. Inset shows sample preparation in a plastic tube, testing in Universal Testing Machine, a crack that developed in the soil sample after testing, and a typical stress–strain graph with the arrow indicating the peak compressive stress.

mound is characterized by high humidity (approx. 98%), high $CO_2$ (up to 4%) and with dampened temperature fluctuations compared to the exterior [9]. We did not attempt to mimic these conditions.

## 3.4. Self-weight consolidation of soil at different moisture contents

Highest peak compressive stress was recorded when the soil was moulded at 30% moisture content and allowed to undergo self-weight consolidation (figure 3a; electronic supplementary material, tables S2 and S3). Self-weight consolidation is the consolidation of slurried deposits under their own weight [44]. The peak compressive stress at this moisture content closely reflected the peak compressive stress of termite mound soil [32] and was significantly different from soil moulded at 40%, 50% and 60% moisture contents (ANOVA peak compressive stress ~ initial soil moisture content: $F_{3,16} = 16.65$, $p \ll 0.001$) suggesting that self-weight consolidation at 30% moisture content is sufficient to impart final achieved *in situ* strength to the mound soil. The peak compressive stress of samples moulded at 15% and 20% moisture contents and subsequently mechanically compacted were not significantly different from each other (ANOVA: peak compressive stress ~ initial soil moisture content: $F_{1,10} = 3.169$, $p = 0.10$) but were lower than samples moulded at 30% and allowed to undergo self-weight consolidation. The dry density of these samples and their accompanying peak compressive stress values are depicted in the electronic supplementary material, table S2 and figure 3a.

A cursory observation of surface features of samples prepared at 15%, 20%, 40%, 50% and 60% initial moisture contents shows different particle arrangements from those moulded at 30% moisture (figure 4). While no distinguishing features or quantification of the structure was possible in these experiments, the soils prepared at 30% water content appear to show a far more homogeneous structure with a packing reminiscent of *in situ* mound soil (figure 4). In samples prepared between 40–60% moisture, the particles are discrete and do not form a homogeneous structure like those prepared at 30% moisture content (figure 4).

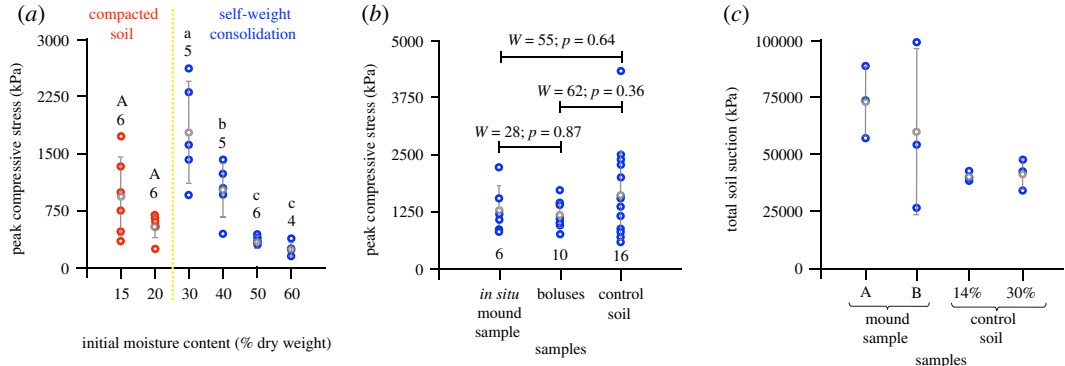

**Figure 3.** Role of moisture in soil strength. (*a*) Peak compressive strength of soil after compaction or self-weight consolidation at different moisture contents and drying. Blue and red circles indicate peak compressive stress of individual samples, grey circles indicate means, and error bars indicate standard deviations. Numbers above circles indicates sample sizes. Capital and small letters indicate significant differences obtained from Tukey's HSD tests performed within the compacted soil and self-weight consolidation groups. (*b*) Strength of *in situ* mound soil samples, boluses and control soil samples prepared at 30–33% moisture content after self-weight consolidation and drying. Blue circles indicate peak compressive stress of individual samples, grey circles indicate means, and error bars indicate standard deviations. Numbers below circles indicate sample sizes. Wilcoxon signed-ranked tests. (*c*) Total soil suction of cored out *in situ* mound soil samples and control soil samples prepared at 14% and 30% moisture content. Blue circles indicate peak compressive stress of individual samples, grey circles indicate means, and error bars indicate standard deviations. *n* = 3 for all sample groups.

## 3.5. Contribution of termite secretions towards strength of aggregated soil

Peak compressive stress of the *in situ* mound soil and of the control soil were not significantly different (Wilcoxon signed-ranked test: $W = 55$, $p = 0.64$; figure 3*b*). Similarly, the peak compressive stress of moulded soil made with boluses under self-weight consolidation was not significantly different from moulds prepared with control soil (Wilcoxon signed-ranked test: $W = 62$, $p = 0.36$; figure 3*b*). Bolus moulds prepared also had the same peak compressive stress as the *in situ* mound soil (Wilcoxon signed-ranked test: $W = 28$, $p = 0.87$; figure 3*b*).

These results corroborate our previous observations described in §§3.3 and 3.4 that self-weight consolidation at 30% moisture content under laboratory conditions is sufficient to impart strength at par with the mound *in situ* soil strength; however, in nature, termite mounds undergo various phenomena such as densification owing to gravity, repeated cycles of wetting and drying, and swelling and shrinking (i.e. soil stress history) which can alter the strength of these samples. These results do not rule out the possibility that termite secretions added during bolus construction contribute to the long-term stability of mounds.

## 3.6. Total soil suction measurement

Soil suction values were higher for the samples extracted from occupied termite mounds than for samples prepared in the laboratory with control soil (figure 3*c*). The relationship between the amount of water in soil and soil suction is defined by the soil–water characteristic curve [15]. According to the soil–water characteristic curve, soil saturated with water has zero soil suction; the soil suction (and the corresponding soil strength) increases with the decrease in soil moisture [15]. Upon further drying, air enters the soil pores [15]. The air-entry value defines the point at which air enters the largest pores of the soil [15]. The shear strength of soil increases linearly up to the air-entry value, after which the relationship becomes nonlinear [15]. The capillary pressure at air entry in the soil is given by the equation:

$$P_c = \frac{2\sigma_{aw}\cos\Theta}{r_p},$$

where $\sigma_{aw}$ is the air–water interfacial tension (0.0728 N m$^{-1}$ at 0°C for pure water), $\Theta$ is the contact angle between air and soil and $r_p$ is the tube radius (radius of capillary formed in the soil = average pore diameter of termite mound soil was taken as 0.53 mm for this calculation [39]). For most soils, the contact angle is close to zero [18]. From this equation and the obtained values, the capillary pressure at air entry for termite mound soil is 549.434 N m$^{-2}$. The total soil suction values obtained in this study

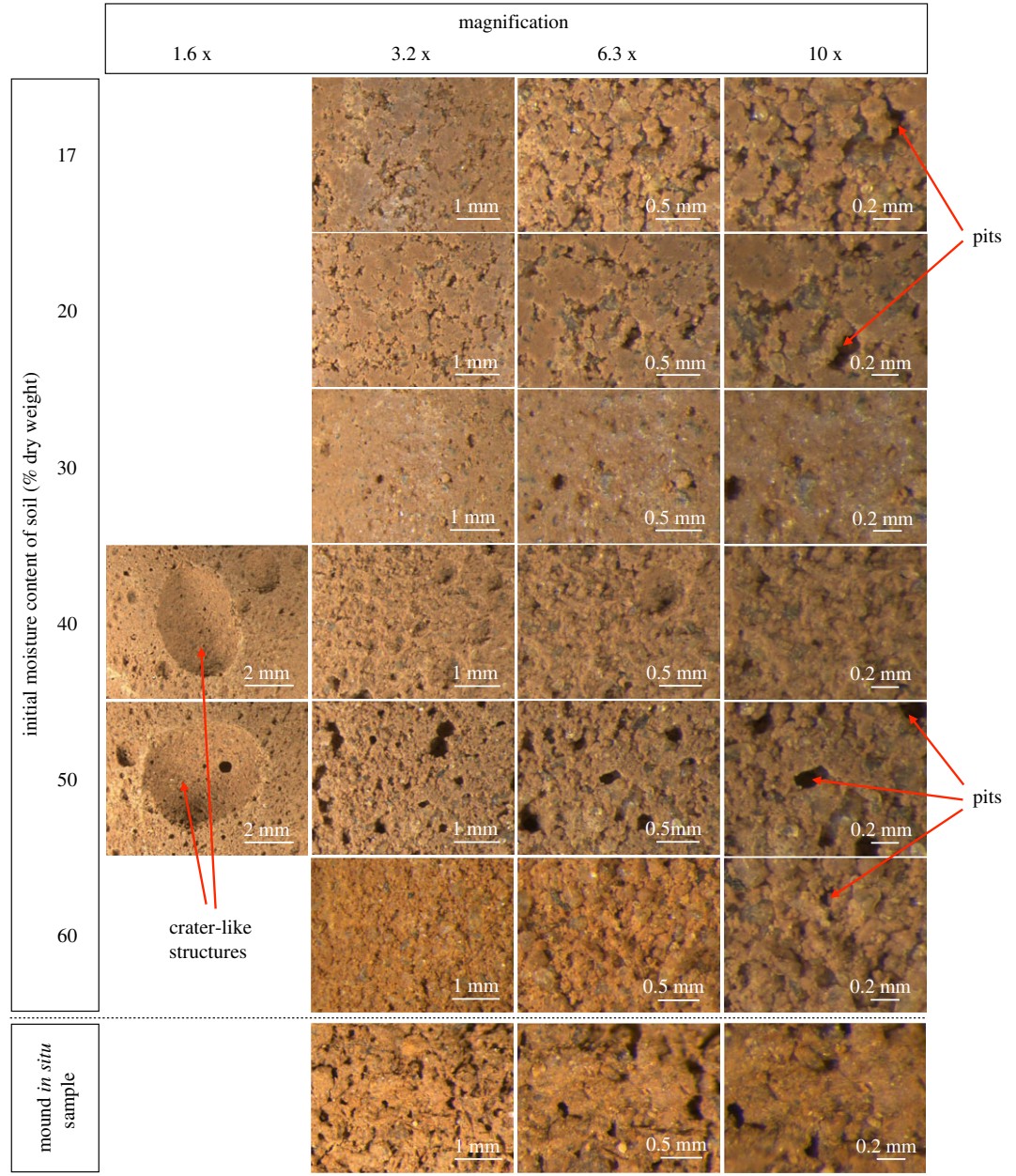

**Figure 4.** Images of samples moulded with 17%, 20%, 30%, 40%, 50% and 60% water each at magnifications 1.6 x , 3.2 x , 6.3 x and 10 x . Crater-like structures were observed in samples moulded with 40% and 50% water.

(figure 3c), therefore, fall in the nonlinear range of the soil–water characteristic curve [15], and their strength correlates may therefore be difficult to compare across samples. While our experiments do not conclusively rule out the possibility of enhanced strength owing to termite secretions, we conjecture that the mechanical strength of the termite mound is largely owing to drying, i.e. soil suction brought forth in the presence of moisture.

## 3.7. Weathering resistance of soil with and without termite manipulation

Resistance to weathering by water was the highest for samples extracted from the abandoned mound followed by samples extracted from occupied mounds and then by control soil samples, indicating that manipulation by termites imparts higher weathering resistance to soil (figure 5). Even after crushing and reconstitution, the mound soil samples retained higher weathering resistance, suggesting that termite secretions added to the soil help in adhering soil particles together and reduce weathering by rain. These additives were stable even at temperatures as high as 80°C because there was no reduction in weathering resistance after drying at this temperature (figure 5). Control soil samples

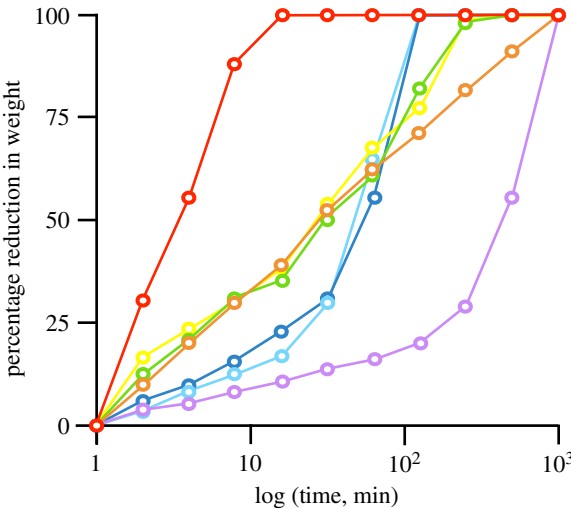

- control soil made at 30% moisture: initially dried at 80°C and after every cycle
- occupied mound soil: initially dried at 80°C and after every cycle
- occupied mound soil: not dried initially; dried after every cycle at 80°C
- occupied mound soil: not dried initially; dried after every cycle at 37.5°C
- reconstituted mound soil at 30% moisture and self-weight consolidation: initially dried at 80°C and after every cycle
- reconstituted mound soil at 17% moisture at *in situ* density: initially dried at 80°C and after every cycle
- sample from abandoned mound: initially dried at 80°C and after every cycle

**Figure 5.** Weathering of mound soil, reconstituted mound soil and control soil during repeated cycles of wetting and drying. Open circles represent averages of six values.

prepared at 30°C as in previous experiments and that demonstrated compressive strength comparable to that of *in situ* mound samples showed the least resistance to weathering; this suggests that weathering resistance and compressive strength are conferred by different aspects of termite soil manipulation. Samples extracted from the lowermost slice of the abandoned mound had the highest weathering resistance, possibly because in addition to termite manipulation they had also experienced other forces over time such as densification owing to gravity and cycles of wetting and drying (ageing effects, i.e. more complex soil history, because the lowest portions of the mounds are also probably the oldest) (figure 5).

## 3.8. Integrated factors involved in long-term stability of termite mounds

Our current study shows that termites manipulate soil close to its liquid limit. Zachariah *et al.* [30] describe that termites cannot use completely dry soil for making boluses and some inherent soil moisture is necessary for soil manipulation. Under laboratory conditions, termites can manipulate soil when the inherent moisture content is in the range of 15–60% (percentage dry weight) [30]. This soil, having undergone self-weight consolidation and upon drying, attains strength close to the *in situ* mound soil strength, suggesting that soil moisture at the time of bolus deposition alone is sufficient to impart strength to the mound. Various factors can play a role in this process such as the particle size of soil used, moisture content of soil and the presence of soil organic matter.

Termite mounds are known to be rich in finer fractions of soil [32]. Study with bidispersed (particles of two sizes) suspension of silica spheres in water shows that as water evaporates, the receding contact line of water drags smaller particles near the larger particles, forming size-segregated aggregates where the large particles are ringed by smaller particles which in turn are ringed by even smaller particles [45]. After water evaporation solid bridges are formed, connecting larger particles to the substrate and to each other forming aggregates [45]. This brings particles within the range of van der Waals interactions which are enhanced by dehydration [45]. The aggregates formed in this manner were stable when subjected to rewetting [45]. This is similar to the case of termite mounds where a range of particle sizes [32], in the presence of moisture [30], is used by termites for construction. Removal of finer particles leads to aggregates that disintegrate upon rewetting, suggesting that polydispersity (presence of a range of

particle sizes) is crucial for solid bridge formation, and the presence of a large number of fine particles increases the surface area thereby increasing attractive van der Waals forces [45]. Moreover, higher polydispersity is expected to enhance the packing configuration, i.e. to bring forth a more closely packed structure between soil particles in the presence of water as suggested by Fall *et al.* [46], facilitating greater consolidation of soil in this case. Therefore, the use of a range of particle sizes in the presence of a suitable amount of moisture in mound construction can facilitate the attained strength of the termite mound.

In our experiments we found that highest density and thereby strength was attained when soil was mixed with water close to its liquid limit (electronic supplementary material, table S2). In polydispersed sand, the addition of a small amount of water reduces the interparticle friction by up to 40% compared to dry sand owing to the formation of liquid-filled pores and liquid bridges around the particle contact points (funicular regime) [46]. The addition of excess water increases the dynamic friction coefficient of sand [46]. Moreover, with increase in the polydispersity of sand, the drop in friction coefficient increases [46]. We expect that similar reduction in dynamic friction coefficient in the mound soil owing to the presence of water around the liquid limit (30% dry weight for the red soil used) will facilitate slippage of soil particles past each other and lead to tight packing.

Organic matter is known to impart high plasticity and high compressibility to soil [40]. Our present study shows that boluses made with burnt soil have lower strength and crumble easily, suggesting the importance of organic matter, in addition to the role of particle size and initial soil moisture content, in imparting strength to the soil. We did not examine the organic content of red soil but according to the literature it is around 4–5% [32]. A preliminary examination of the content of termite salivary secretions (data not shown) was inconclusive in terms of identifying any particular groups of compounds that could be responsible for the cohesion of soil particles, thereby enhancing the strength of the soil constructions.

We conjecture that termite mound strength is specifically achieved as a consequence of matrix suction and bidispersity of the bolus deposition (mound construction using boluses of two different sizes that are packed together) [30] which results in a monolithic, densely packed structure of the mound.

In this study, we also found that termite mound soil shows exceptional resistance to weathering by water. The samples extracted from an abandoned mound had the highest resistance to weathering followed by the soil from occupied mounds followed by control soil. The sample from an abandoned mound must have contained termite secretions as additives, and probably experienced a long and complex stress history owing to frequent wetting–drying cycles and also re-building and movement of soil by termites. In the case of intact and reconstituted samples from occupied mounds, the presence of termite secretions imparted weathering resistance. The control soil had neither termite secretions nor stress history; therefore, it had the least resistance towards weathering. Termite secretions as additives therefore imparted very high weathering resistance to the samples. The abandoned mound samples were stored in the laboratory for 6 years before carrying out these experiments. This suggests that despite such long storage times the overall structure of the mound was retained intact. This can be one of the reasons why the remains of termite mounds can last for centuries [31]. In natural settings, frequent repair and remodelling of the mounds may alleviate the impact of weathering by rain and wind. Unlike weathering caused by biological agents [28], termite secretions appear to resist weathering in termite mound soil, possibly by interacting with soil organic matter. Currently, it is not known how termite secretions interact with soil organic matter and retard various forms of chemical weathering (such as dissolution, hydration, carbonation, hydrolysis, redox reactions, cation exchange). It is also not known if termite secretions increase/decrease interparticle friction or provide any additional structure or rigidity to soil fabric.

Biocementation plays an important role in various animal-built structures. It allows animals to use a large amount of collected materials (exogenous origin) and to employ a very small amount of secretion for construction (endogenous origin). This is advantageous because the energy spent in collecting materials and secreting biocement can be optimized [47]. In the case of termite mounds, the usage of abundantly available exogenous material (soil and water) with the addition of some amount of endogenous material (termite secretions) can provide a possible mechanism for minimizing energy expenditure during mound construction.

Currently, it is not known if convergent evolution has led to the widespread use of soil moisture as a means of providing strength to the soil in animal-built structures such as the nests of potter wasps and cliff and barn swallows. If so, then such a strategy, coupled with construction site selection shielded from direct rain [2,48], can possibly eliminate the need for endogenous additives for achieving strength and weathering resistance of their built nest, leading to a reduction in the energetic cost of construction.

In this study, we provide, to our knowledge, the first empirical evidence that soil moisture at the time of bolus deposition alone is sufficient to impart strength to the mound after drying, suggesting an efficient mechanism of minimizing energy expenditure in mound construction by reducing the need for endogenous additives for attaining required strength of the construction. However, weathering resistance does require the addition of termite secretions, the exact mechanism of action being yet unknown. If termites secrete a small quantity of an additive chemical cocktail that acts on the soil organic matter converting it into a product that imparts weathering resistance, then the energetic cost of this mechanism would still be lower than a case where termites secrete separate cementing agents and those that impart weathering resistance. Earlier we had shown that the presence of moisture is necessary for soil manipulation by termites [30]. Ground water [49], the water of metabolic origin [50] and water in the form of termite salivary glue [51] have been suggested to play a role in mound construction in other termite species. However, the source of water used for construction in *O. obesus* mounds is not known. This, coupled with our present results, indicate that the appropriate amount of moisture is necessary and sufficient for attaining strength in termite mounds. By manipulating soil at its liquid limit, termites make mounds with optimum strength, ventilation and high slope stability which also allows remodelling over time with colony growth. Our results can have significant implications for the construction of man-made earthen structures.

Data accessibility. Our data are available in the electronic supplementary material Information and deposited at Dryad Digital Repository: https://doi.org/10.5061/dryad.h18931zh1 [52].

Authors' contributions. N.Z., T.G.M. and R.M.B conceived and designed the study; N.Z., T.G.M. and R.M.B wrote the manuscript; N.Z. performed experiments; N.Z. analysed the data; N.Z., T.G.M. and R.M.B interpreted the data. All authors gave final approval for publication.

Competing interests. We declare we have no competing interests.

Funding. Financial support came from the Department of Biotechnology–IISc Partnership Programme; Department of Science and Technology-FIST; Council of Scientific and Industrial Research; Ministry of Environment, Forests and Climate Change, Government of India and the CeNSE facilities at IISc funded by Department of Information Technology.

Acknowledgements. We thank Aditi Vijayan, S. Thejaswini, Surbhi Chouhan and Kruthika Sen for help in performing experiments and Yathiraj Ganesh for help with fieldwork. We are grateful to Akshata G. Bhat and S. Anantu for help with imaging of soil samples, and to Monika Rekapalli, Saurabh Singh, Ramesh Kandasami and an anonymous reviewer, who provided comments that substantially improved the manuscript.

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
