## [Reviewer comments · Royal Society Open Science]

Review History

RSOS-200485.R0 (Original submission)

Review form: Reviewer 1 (David Hu)

Is the manuscript scientifically sound in its present form?

Yes

Are the interpretations and conclusions justified by the results?

No

Is the language acceptable?

Yes

Do you have any ethical concerns with this paper?

Yes

Have you any concerns about statistical analyses in this paper?

No

Recommendation?

Accept with minor revision (please list in comments)

Comments to the Author(s)

The authors study the strength of termite mounds. Their main result is that termites tend to choose soil with a 30 percent moisture content, corresponding to soil with highest strength (here measured as peak compressive stress). I like that this was found purely from observing the choices made by termites. However, the authors should improve their presentation to make this work more stand alone. They assume that the reader is familiar with the literature and their previous papers. Many terms are not defined, making it difficult to understand their exact procedures. Once these are defined and clarified, this paper should be publishable.

The methods for measuring peak compressive stress should be given.

line 49 soil suction should be defined. Its not clear what those experiments are.

106, line 152, 265: here the author references an unpublished study but doing so does not provide the evidence needed. These lines should be removed.

134 the results of the previous experiment"

141 this line is unclear. Why at the purpose of air drying and then oven drying? The air drying seems redundant. Also, the relation of strength and density is not clear.

203 weathering of soil should be defined. The authors are putting their samples in water, but its not clear what they are measuring. Its also not clear why there are 6 categories of soil being measured here.

247 the plastic region should be defined a few lines earlier to avoid confusion. plastic and liquid limits should be defined.

263 it seems the soil becomes strongest at 12 days of drying. What is going on during these 12 days?

274 self weight consolidation should be defined.

301 the authors often use the terms "previous observations" but the writing is not clear. They should be explicit about what measurements.

316 how are the authors measuring suction pressure, reported in fig 4d? Clearly, the pore size in soil will pull in water by capillarity, but how does this relate to strength? The section on 310 on the soil-suction curve should be explained.

330 its strange that the abandoned mounds were more resistant to weathering than currently occupied mounds. Is there any aging effect that happens over many years? The authors should give some hypothesis.

348 how do the termites control the wetness of the bolus. Do they mix with saliva, dip in water? It would be nice to know a little of this process. How much water do they need to add to typical soil to make it the desired moisture content.

381 the comment on the burnt soil refers to data, but I don't see it in the figures.

Fig 1. The normalization coefficient should be given in the caption.

Fig S1 helps the reader understand the experiments and a version might be given in the main text.

There are a number of typos in the references.

Decision letter (RSOS-200485.R0)

Dear Dr Borges

On behalf of the Editors, I am pleased to inform you that your Manuscript RSOS-200485 entitled "Moisture alone is sufficient to impart strength but not weathering resistance to termite mound soil" has been accepted for publication in Royal Society Open Science subject to minor revision in accordance with the referee suggestions. Please find the referees' comments at the end of this email.

The reviewers and handling editors have recommended publication, but also suggest some minor revisions to your manuscript. Therefore, I invite you to respond to the comments and revise your manuscript.

- Ethics statement

- Data accessibility

<http://datadryad.org/submit?journalID=RSOS&manu=RSOS-200485>

- Competing interests

- Authors' contributions

- Acknowledgements

- Funding statement

Because the schedule for publication is very tight, it is a condition of publication that you submit the revised version of your manuscript before 19-Jun-2020. Please note that the revision deadline will expire at 00.00am on this date. If you do not think you will be able to meet this date please let me know immediately.

- 1) A text file of the manuscript (tex, txt, rtf, docx or doc), references, tables (including captions) and figure captions. Do not upload a PDF as your "Main Document";
- 2) A separate electronic file of each figure (EPS or print-quality PDF preferred (either format should be produced directly from original creation package), or original software format);
- 3) Included a 100 word media summary of your paper when requested at submission. Please ensure you have entered correct contact details (email, institution and telephone) in your user account;
- 4) Included the raw data to support the claims made in your paper. You can either include your data as electronic supplementary material or upload to a repository and include the relevant doi within your manuscript. Make sure it is clear in your data accessibility statement how the data can be accessed;

5) All supplementary materials accompanying an accepted article will be treated as in their final form. Note that the Royal Society will neither edit nor typeset supplementary material and it will be hosted as provided. Please ensure that the supplementary material includes the paper details where possible (authors, article title, journal name).

If your manuscript is newly submitted and subsequently accepted for publication, you will be asked to pay the article processing charge, unless you request a waiver and this is approved by Royal Society Publishing. You can find out more about the charges at <https://royalsocietypublishing.org/rsos/charges>. Should you have any queries, please contact openscience@royalsociety.org.

on behalf of Kevin Padian (Subject Editor)
openscience@royalsociety.org

Reviewer comments to Author:
Reviewer: 1

Comments to the Author(s)

The authors study the strength of termite mounds. Their main result is that termites tend to choose soil with a 30 percent moisture content, corresponding to soil with highest strength (here measured as peak compressive stress). I like that this was found purely from observing the choices made by termites. However, the authors should improve their presentation to make this work more stand alone. They assume that the reader is familiar with their literature and their previous papers. Many terms are not defined, making it difficult to understand their exact procedures. Once these are defined and clarified, this paper should be publishable.

The methods for measuring peak compressive stress should be given.

line 49 soil suction should be defined. Its not clear what those experiments are.

106, line 152, 265: here the author references an unpublished study but doing so does not provide the evidence needed. These lines should be removed.

134 the results of the previous experiment"

141 this line is unclear. Why at the purpose of air drying and then oven drying? The air drying seems redundant. Also, the relation of strength and density is not clear.

203 weathering of soil should be defined. The authors are putting their samples in water, but its not clear what they are measuring. Its also not clear why there are 6 categories of soil being measured here.

247 the plastic region should be defined a few lines earlier to avoid confusion. plastic and liquid limits should be defined.

263 it seems the soil becomes strongest at 12 days of drying. What is going on during these 12 days?

274 self weight consolidation should be defined.

301 the authors often use the terms "previous observations" but the writing is not clear. They should be explicit about what measurements.

316 how are the authors measuring suction pressure, reported in fig 4d? Clearly, the pore size in soil will pull in water by capillarity, but how does this relate to strength? The section on 310 on the soil-suction curve should be explained.

330 its strange that the abandoned mounds were more resistant to weathering than currently occupied mounds. Is there any aging effect that happens over many years? The authors should give some hypothesis.

348 how do the termites control the wetness of the bolus. Do they mix with saliva, dip in water? It would be nice to know a little of this process. How much water do they need to add to typical soil to make it the desired moisture content.

381 the comment on the burnt soil refers to data, but I don't see it in the figures.

Fig 1. The normalization coefficient should be given in the caption.

Fig S1 helps the reader understand the experiments and a version might be given in the main text.

There are a number of tyoes in the references.

Author's Response to Decision Letter for (RSOS-200485.R0)

See Appendix A.

Decision letter (RSOS-200485.R1)

Dear Dr Borges,

It is a pleasure to accept your manuscript entitled "Moisture alone is sufficient to impart strength but not weathering resistance to termite mound soil" in its current form for publication in Royal Society Open Science. The comments of the reviewer(s) who reviewed your manuscript are included at the foot of this letter.

on behalf of Kevin Padian (Subject Editor)
openscience@royalsociety.org

Appendix A

RESPONSE TO REVIEWERS

We thank the reviewers for their insightful comments. We present the response to the reviewers in blue font. The corresponding changes in the manuscript are highlighted in red and are annotated accordingly here.

Reviewer #1 (Technical Comments to the Author):

The authors study the strength of termite mounds. Their main result is that termites tend to choose soil with a 30 percent moisture content, corresponding to soil with highest strength (here measured as peak compressive stress). I like that this was found purely from observing the choices made by termites. However, the authors should improve their presentation to make this work more stand alone. They assume that the reader is familiar with the literature and their previous papers. Many terms are not defined, making it difficult to understand their exact procedures. Once these are defined and clarified, this paper should be publishable.

The methods for measuring peak compressive stress should be given.

Authors' response: We have described the method for measuring peak compressive stress in the revised manuscript (Lines 111–114).

line 49 soil suction should be defined. It's not clear what those experiments are.

Authors' response: We have defined soil suction in the revised manuscript. Section 2.8 and Section 3.6 refer to these measurements (Lines 49–56).

106, line 152, 265: here the author references an unpublished study but doing so does not provide the evidence needed. These lines should be removed.

Authors' response: The unpublished study we had referred to in the previous version of the manuscript has recently been accepted in the journal Scientific Reports. We have cited this accepted paper in the revised manuscript (Lines 169, 292) as Reference No. 39 [Zachariah N, Singh S, Murthy TG, Borges RM. 2020 Bi-layered architecture facilitates high strength and ventilation in nest mounds of fungus-farming termites. Sci. Rep. (Accepted; DOI not yet received)].

134 the results of the previous experiment"

Authors' response: We have made changes to this sentence and have indicated that it is referring to sections 2.4 and 3.2 of the manuscript (Line 146).

141 this line is unclear. What is the purpose of air drying and then oven drying? The air drying seems redundant. Also, the relation of strength and density is not clear.

Authors' response: It is well known in the engineering literature that the density of soil affects its strength — higher the density higher is the strength. Air drying allowed self-weight consolidation of soil leading to arrangement and interlocking of soil particles. Air drying of soil samples was an attempt at mimicking the natural process of drying of soil in the termite mound. The remnant moisture in the soil sample was removed by oven drying in order to ensure that it did not affect the process of compressive strength testing (Lines 159–163).

203 weathering of soil should be defined. The authors are putting their samples in water, but its not clear what they are measuring. Its also not clear why there are 6 categories of soil being measured here.

Authors' response: We have defined weathering in the revised manuscript (Lines 67–70). By putting soil samples in water we are measuring the weathering of soil i.e. dissolution of interparticle bonds in soil. This method was adapted from Kandasami et al., (2016) for measuring weathering resistance. In this experiment we were trying to decipher the effect of termite secretions, mound *in-situ* particle arrangement, and soil history (ageing effects) on weathering resistance of soil (Lines 225–240).

247 the plastic region should be defined a few lines earlier to avoid confusion. plastic and liquid limits should be defined.

Authors' response: We have defined this in the revised manuscript (Lines 266–271).

263 it seems the soil becomes strongest at 12 days of drying. What is going on during these 12 days?

Authors' response: During these 12 days, the sedimentation of soil particles and evaporation of water from the interparticle spaces would have led to the interlocking of soil particles and the formation of capillary bridges between the soil particles. We now mention this in the revised manuscript (Lines 292–294).

274 self weight consolidation should be defined.

Authors' response: We have defined self-weight consolidation in the revised manuscript (Lines 303–304).

301 the authors often use the terms "previous observations" but the writing is not clear. They should be explicit about what measurements.

Authors' response: We have indicated in our revised manuscript that “previous observations” refers to section 3.3 and section 3.4 of the manuscript (Line 331).

316 how are the authors measuring suction pressure, reported in fig 4d? Clearly, the pore size in soil will pull in water by capillarity, but how does this relate to strength? The section on 310 on the soil-suction curve should be explained.

Authors' response: We measured the suction pressure by exactly following the protocol described in ASTM International. (see following cited reference)[2019 ASTM D5298 - 16 Standard test method for measurement of soil potential (suction) using filter paper (ASTM International, West Conshohocken, PA)]

In nature, the pores between the soil particles can be filled with moisture — higher the moisture, lower is the soil strength. As the soil dries, the amount of moisture between the soil particles reduces. At lower moisture content, the soil particles are held together by the surface tension of water. Soil suction is conceptualised as stresses arising from interparticle cementation, van der Waals attraction, double-layer repulsion and capillary stresses. Soil suction in practical terms is a measure of the affinity of soil to retain water and can provide information on soil parameters that are influenced by the soil water; for example, volume change, deformation, and strength characteristics of the soil. The ASTM protocol followed for measuring soil suction requires the soil sample to be kept in an airtight container with a Whatman filter paper No. 42 physically separated from each other by a wire mesh. The setup is to be left undisturbed for 7 days. During this time, water will evaporate from the soil and will be absorbed by the Whatman filter paper. At the end of the experiment the moisture content in the filter paper is measured and the soil suction calculated as per ASTM protocol. The higher the moisture content in the filter paper the lower is the soil suction which means that with more water between the soil particles, there is lower strength in the soil. Since this

entire experimental procedure is described in detail in the cited reference, therefore in the interest of avoiding repetition we have not described it separately in the manuscript.

We have described the soil–water characteristic curve in the revised manuscript.

(Lines 342–344).

330 its strange that the abandoned mounds were more resistant to weathering than currently occupied mounds. Is there any aging effect that happens over many years? The authors should give some hypothesis.

Authors' response: We had already provided a hypothesis in the manuscript (Lines in original manuscript 340–344 and 395–397; Lines in revised manuscript: 372–376 and 430–432).

348 how do the termites control the wetness of the bolus. Do they mix with saliva, dip in water? It would be nice to know a little of this process. How much water do they need to add to typical soil to make it the desired moisture content.

Authors' response: We have described in our previous research paper Zachariah et al., (2017) that termites cannot use completely dry soil for making boluses and some inherent soil moisture is necessary for soil manipulation. Under laboratory conditions, termites can manipulate soil when the inherent moisture content is in the range of 15–60% (percentage dry weight). In addition to this, termites also mix the soil with their saliva. However, it is not known if termites reduce the moisture content of soil in nature for bringing it close to the liquid limit (and this is an extremely unlikely event). Therefore, most likely termites collect soil from their surroundings which is below the liquid limit and mix it with their secretions to bring it to the liquid limit before depositing the bolus. We do not know this for sure and we are unable to speculate, but we add more information on this process (Lines 378–382).

381 the comment on the burnt soil refers to data, but I don't see it in the figurese.

Authors' response: We have described this in section 2.2 “Burnt soil samples (boluses and soil unmanipulated by termites) were extremely fragile and, upon packing in cylindrical moulds and subjected to drying, would crumble. Therefore, all further experiments with soil were conducted with red soil (soil containing organic matter).” Since soil devoid of organic matter crumbled in upon drying, its strength was zero. (Lines in original manuscript: 102–105; Lines in revised manuscript: 115–118).

Fig 1. The normalization coefficient should be given in the caption.

Authors' response: We have made the suggested change in the figure legend (Line 648).

Fig S1 helps the reader understand the experiments and a version might be given in the main text.

Authors' response: Thank you for the suggestion. In the revised manuscript we have included a version of Fig. S1 in Fig. 2 of the main text as an inset. In the same inset we have also included an image of the sample being tested in a Universal Testing Machine, a crack that developed in the soil sample after testing, and a typical stress–strain graph for easy understanding (Lines 655–657).

There are a number of typos in the references.

Authors' response: We have corrected the typos in the references.